# Antibacterial Aerogels-Based Membranes by Customized Colloidal Functionalization of TEMPO-Oxidized Cellulose Nanofibers Incorporating CuO

**DOI:** 10.3390/bioengineering10111312

**Published:** 2023-11-14

**Authors:** Elena Usala, Eduardo Espinosa, Wasim El Arfaoui, Ramón Morcillo-Martín, Begoña Ferrari, Zoilo González

**Affiliations:** 1BioPren Group (RNM940), Chemical Engineering Department, Instituto Químico Para la Energía y el Medioambiente (IQUEMA), Faculty of Science, Universidad de Córdoba (UCO), 14014 Córdoba, Spain; z12ususe@uco.es (E.U.); eduardo.espinosa@uco.es (E.E.); t62momar@uco.es (R.M.-M.); 2Unidad Asociada CSIC-UCO, Fabricación Aditiva de Materiales Compuestos Basados en Celulosa Funcionalizada, Obtenida de Residuos de Biomasa, 14014 Córdoba, Spain; bferrari@icv.csic.es; 3Instituto de Cerámica y Vidrio, Consejo Superior de Investigaciones Científicas (CSIC), Campus de Cantoblanco, c/Kelsen 5, 28049 Madrid, Spain

**Keywords:** colloidal heterostructures, functional aerogels, cellulose nanofibers, CuO nanoparticles, antibacterial membranes

## Abstract

An innovative colloidal approach is proposed here to carry out the customized functionalization of TEMPO-Oxidized Cellulose Nanofibers (CNF) incorporating non-noble inorganic nanoparticles. A heterocoagulation process is applied between the delignified CNF and as-synthetized CuO nanoparticles (CuO NPs) to formulate mixtures which are used in the preparation of aerogels with antibacterial effect, which could be used to manufacture membranes, filters, foams, etc. The involved components of formulated blending, CNF and CuO NPs, were individually obtained by using a biorefinery strategy for agricultural waste valorization, together with an optimized chemical precipitation, assisted by ultrasounds. The optimization of synthesis parameters for CuO NPs has avoided the presence of undesirable species, which usually requires later thermal treatment with associated costs. The aerogels-based structure, obtained by conventional freeze-drying, acted as 3D support for CuO NPs, providing a good dispersion within the cross-linked structure of the nanocellulose and facilitating direct contact of the antibacterial phase against undesirable microorganisms. All samples showed a positive response against *Escherichia coli* and *Staphylococcus aureus*. An increase of the antibacterial response of the aerogels, measured by agar disk diffusion test, has been observed with the increase of CuO NPs incorporated, obtaining the width of the antimicrobial “halo” (nw_halo_) from 0 to 0.6 and 0.35 for *S. aureus* and *E. coli*, respectively. Furthermore, the aerogels have been able to deactivate *S. aureus* and *E. coli* in less than 5 h when the antibacterial assays have been analyzed by a broth dilution method. From CNF-50CuO samples, an overlap in the nanoparticle effect produced a decrease of the antimicrobial kinetic.

## 1. Introduction

The application of polymer-based porous structures (membranes, filters, foams) in wastewater treatment or in the concentration of targeted chemicals from aqueous solution is widespread. Nevertheless, the longstanding biofouling issue on structure surfaces (internal and external) has garnered enormous attention within the scientific and industrial fields. Motivated by the recent development of nanocomposite materials, the integration of antimicrobial nanophases within polymeric structures is emerging as a promising strategy to mitigate biofouling [1,2].

On the other hand, the environmental problems and the limited non-renewable resources are encouraging the search for alternative materials. The need to reduce the exploitation in fossil fuels drives the alternative use of other eco-friendly resources. Concerns about sustainability have attracted great interest in bioproducts and renewable materials, from biorefinery, as emerging solutions to satisfy a range of technological challenges [3].

In this sense, the manufacturing of composite materials, incorporating naturally derived polymers coming from valorization processes, has emerged as a promising class of engineering materials, providing new prospects for modern technology. The implementation of this type of energy-efficient is vital to develop the long-term bioeconomy concept.

Among all exploitation possibilities, the use of nanocellulose-based materials is very promising for their biocompatibility and unique properties such as low density, biodegradability, abundance, high-functionalization possibilities, and their vast applicability [4,5]. In addition to possessing the inherent properties of a polymer obtained from natural resources, cellulose nanofibers (CNF) can be combined with other functional components to gain certain added value. The combination of two or more constituents offers better properties than the individual components. In particular, the incorporation of metal oxide nanoparticles (MONPs) is considered an efficient approach to obtain potential features in the fields catalysis, water purification, etc. [6]. Controlling the concentration and physicochemical characteristics (size, morphology, specific surface area, porosity, etc.) of the inorganic phases, the functionalities of composites materials can be tailored and improved. In this sense, the employment of ultrasound systems to assist the syntheses of MONPs is today considered an interesting way to regulate nucleation and growth during their processes of formation. The impact of ultrasound on crystallographic, textural, and functional properties of MONPs has been intensively investigated in the last decades, revealing that final performances are influenced by the presence of cavitation bubbles which induces the ultrasonic waves [7].

In the last few years, many hybrid nanocellulose/MONPs formulations have been proposed to prepare functional composite as thin films, coated-papers, and other planar format [8,9]. Additionally, the manufacturing of three-dimensional networks, such as hydrogels, foams, crumpled shapes, and especially aerogels, have attracted special attention as potential frameworks or supporting matrixes of MONPs. They possess interesting characteristics such as immobilization ability, high specific surface area, structural integrity, mechanical robustness, high porosity, and easy separation of the material from reactions media [10,11].

The incorporation of copper oxide nanoparticles (CuO NPs) in nanohybrid structures has been particularly studied in the last years due to the broad spectrum of applications. One of the most important characteristics is their antimicrobial activity against bacteria, fungi, viruses, and algae, which is extremely interesting for its use as polymer nanocomposite [12]. Moreover, the CuO can be considered a cheaper alternative than the conventional use of noble metal nanoparticles. Their inherent properties also interact closely with the microorganism membranes and exhibit competitive antimicrobial effects [13]. Some authors have already worked on the preparation of nanocellulose/CuO composites. For example Heidari, H et al. carried out an in situ synthesis of CuO on bacterial cellulose and measured the antibacterial effects against strains of *Klebsiella pneumonia* and *Staphylococcus aureus* [13]. Almasi, H. et al. compared the efficiency against *E. coli*, *P. aeruginosa*, and *L. monocytogenes* of bacterial cellulose nanofibers and chitosan nanofibers containing precipitated CuO [14]. On the other hand, Shankar et al. demonstrated that the characteristics of CuO, synthesized in the presence of cellulose, were additionally affected by the type of cellulose used. They found that the source of cellulose influences the shape, size, and efficiency of cellulose–CuO composite [15].

In all cases, the incorporation of inorganic NPs, including that mentioned with CuO, has been carried by using conventional wet chemical methods such as sol–gel, co-precipitation, and hydrothermal synthesis [16,17], where certain relevant processing parameters have not been considered in detail.

This is the reason why the application of an alternative colloidal approach could become a design guide to achieve major control over manufacturing processes and therefore produce more efficient systems. The development of high homogeneous and well-dispersed components in suspension, as well as the study of their electrostatic interactions, allow for controlling the mixtures in liquid media and obtaining optimum distributions, which are completely relevant for the subsequent shaping processes [18,19].

Therefore, the present study focused on the customized preparation of functional CNF-based aerogels incorporating antibacterial CuO by using a facile colloidal methodology. The formulation of CNF-CuO blends was carried out by a heterocoagulation route described previously with other inorganic NPs. Furthermore, an analysis of the aerogel structure and their functional properties when it is functionalized with different content of CuO was undertaken. All samples have been characterized by XRD, FESEM, and FTIR. In addition, the antimicrobial activities of the aerogels were determined against *Staphylococcus aureus* and *Escherichia coli* by Broth dilution method and Agar Diffusion method.

## 2. Materials and Methods

### 2.1. Preparation of CuO NPs and CNF Suspensions

The CuO nanopowder used in this work was firstly synthetized by a chemical precipitation route, assisted by a high intensity ultrasonic horn for 30 min (SONOPULS HD 4200 BANDELIN using the Ti plate TT213, ø = 13 mm with a power of 150 W, Berlin, Germany). The procedure was carried out by dissolving the appropriate amount of copper salt (0.1 mol·L^−1^ of CuSO_4_·5H_2_O) in 20 mL of deionized water. The reaction was carried out by adding, drop by drop, 20 mL of potassium hydroxide solution of 0.3 mol·L^−1^. The powder obtained was filtered using polymeric filters (ø = 0.22 μm), washed with deionized water, and dried at 60 °C for 24 h. The parameters were adjusted to avoid the presence of undesirable species and the application of later calcination processes. The yield of the synthesis was finally determined according to stoichiometric reaction.

The crystalline structure, interlaminar distance, and crystallite size of the obtained powder was studied by XRD analysis using a D8 Advance diffractometer (Bruker, monochromatic Cu Kα radiation λ = 1.540598 Å, 40 kV; 40 mA; 2 h = 10–80). Nitrogen adsorption/desorption and specific surface area on the synthetized powder were measured at 77 K using an adsorption analyzer (Autosorb-iQ-2 MP/XR, Quantachrome, Madrid, Spain). A thermal characterization of the synthetized powder was also made by thermogravimetric analysis (5 °C min^−1^, T = 25–600 °C, Air atmosphere, TGA/DSC 1 STAR system (Mettler Toledo, Spain)). The size and morphology of the CuO powder was also examined by a FE-SEM JEOL JSM 6300 microscope (Boston, MA, USA).

On the other hand, TEMPO-Oxidized Cellulose nanofibers (CNF) were prepared from the valorization of wheat straw residues by a standard biorefinery procedure described elsewhere [20,21].

In comparison with the CNF obtained by a mechanical procedure, these CNF presented excellent physicochemical properties with a high nanofibrillation yield value (~87%), high cationic demand and carboxyl content (~1210 and ~360 μeq·g^−1^, respectively), and large specific surface area (414 m^2^·g^−1^).

Before blending the components of composite, individual colloidal stabilization analyses of CuO and CNF were made at different pH by zeta potential measurements. The colloidal dispersion of CuO was additionally studied by adsorbing different amounts of a cationic polyelectrolyte as a stabilizer additive. Branched polyethylenimine (PEI, Mw 25,000 Sigma-Aldrich, Madrid, Spain) was used for this purpose. All Zeta potential measurements were made by laser Doppler velocimetry in a ZetasizerNano ZS (Malvern Panalytical, Malvern, UK). The zeta potential evolution of NP with pH was determined for suspensions with concentrations of 0.01 g·L^−1^, using 10^−2^ M KCl as a solvent and inert electrolyte to maintain the ionic strength of the medium. The pH adjustment of the suspensions was carried out by the addition of small quantities of 0.1 M HNO_3_ or KOH and was controlled with a pH probe (Metrohm AG, Madrid, Spain). The zeta potential saturation curve determined for the addition of PEI was measured without the inert electrolyte at pH 8. Before each measurement, NP suspensions were dispersed and homogenized by sonication, using an ultrasonic horn (Sonopuls HD4200, 100 W, Bandelin electronic GmbH & Co. Bandelin, Germany). On the other hand, the homogenization of CNF was made by using a high-shear homogenizer (IKA T18 digital Ultra Turrax, IKA, Staufen, Germany).

### 2.2. Preparation of Nanocellulose/CuO Aerogels

The preparation of the CNF/CuO-PEI structures was carried out by adding the CuO-PEI suspension on the CNF hydrogel, inducing the heterocoagulation phenomenon. Both the surface charge of CNF and the exact amount of CuO-PEI to completely cover the CNF surface were also monitored in terms of zeta potential. The pH of the suspension was around 8 to increase the protonation of amine groups of PEI, favoring the adsorption, and the later interaction with the carboxyl/hydroxyl groups presented in nanofibers.

For this, varying amounts of the stock suspensions of CNF (0.4 wt.%) and CuO-PEI, previously prepared as reference, were combined to obtain mixtures with different percentages (5, 10, 25, 50, and 100 wt.% CuO-PEI on dry weight of CNF. The obtained samples have been named as CNF-XCuO). All blends were cast in PET molds with a 25 mm diameter, frozen for 24 h, and freeze-dried at −85 °C under 0.5 mBar for 72 h in a Lyoquest −85 device.

ATR-FTIR spectra of the compounds was carried out in a Perkin Elmer spectrometer, by mounting the samples onto its diamond cell and pressing with an adjustable screw. IR transmittance was measured in the range 400–4000 cm^−1^ with a resolution of 4 cm^−1^.

The CuO NPs distributions on cellulose-based matrix as well as the microstructural morphology of the resulting drying aerogels were additionally observed with a field emission scanning electron microscope (FESEM, Hitachi S4700, Hitachi Ltd, Ibaraki, Japan). All samples were prepared by using electrically conductive, non-porous carbon tapes suitable for SEM. Moreover, a small amount of conductive carbon slurry was applied between the tape and the aerogel sample to favor their contact. After that, samples were coated with a thin layer of Au by using conventional sputtering equipment.

### 2.3. Antimicrobial Analysis of the CNF/CuO Aerogels

The antimicrobial properties of the produced aerogels were tested against two different microorganisms—the Gram-positive bacteria *S. aureus* and the Gram-negative bacteria *E. coli*.

#### 2.3.1. Agar Disk Diffusion Method

To perform the agar disk diffusion tests, three CNF-XCuO aerogels of each composition were cut into round shapes of approximately 10 mm diameter and were exposed to UV-light for 20 min. Lawns of the target microorganisms (1 × 10^8^ cfu/mL) were performed in the surface of the nutrient agar (NA) plates. The resulting aerogels were placed upon the lawns and kept at 37 °C for 24 h. After 24 h, the means and standard deviations of the normalized widths of the antimicrobial “halos” (nw_halo_) were determined by applying the methodology described elsewhere [22] through the following equation:nw_halo_ = [(d_iz_ − d)/2]/d(1)
where d_iz_ is the diameter of the inhibition zone and d is the disk diameter.

#### 2.3.2. Broth Dilution Method

The antimicrobial activity in liquid medium was also tested for the different CNF-XCuO compositions. The means and the standard deviations were also determined from two aerogel samples of each composition. First, an aliquot of the 40% (*v*/*v*) glycerol−ether frozen stock was cultured in nutrient broth (NB) at 37 °C for 24 h. To obtain cells in the stationary growth phase, bacteria were subcultured at 37 °C for 24 in fresh NB. The bacterial concentration was adjusted to 10^8^ CFU/mL using the McFarland 0.5 standard (0.1–0.08 absorbance at 625 nm). Tubes containing the CNF-XCuO aerogels were initially prepared in NaCl solution (0.85% *w*/*v*). Each test tube was inoculated with 100 µL of the previously standardized solution, obtaining a final concentration of approximately 10^6^ CFU/mL in the tubes. To confirm the initial bacterial concentration, a plate count of the standard tube was carried out. Afterwards, test tubes were incubated at the optimum conditions for each microorganism (37 °C) and, subsequently, samples were taken from each tube at specific time intervals: 0, 1, 2.5, 5, 7, and 24 h. For each tube the appropriate serial dilutions were performed, and 0.1 mL was surface-plated in NA. After the incubation period (24 h at 37 °C), plates were counted. As negative controls, tubes containing CNF-100% aerogels + inoculum (CNF) and inoculum + NaCl solution (Control) were tested.

## 3. Results and Discussion

### 3.1. Synthesis and Characterization of the Involved Components

The manufacturing of the antibacterial aerogels was carried out by functionalizing TEMPO-Oxidized CNF.

On the other hand, previous results showed that the purity, size, and morphology of CuO nanopowders, synthesized through a conventional precipitation reaction, are affected for the type of precursor used [23]. Moreover, a calcination process above 450 °C is required to ensure the crystallinity and purification of the CuO specie [24].

Our work demonstrates that the application of a controlled amount of ultrasonic power in a concrete volume of precursor solution (W cm^−2^ mol^−1^) allowed the instantaneous formation of a pure CuO monoclinic phase in the absence of thermal treatments. The effect of the applied ultrasounds leads to higher Gibbs free energy values avoiding the presence of intermediated precipitates, such as Cu(OH)_2_ or Cu_2_O, which are normally obtained applying lower calcination temperatures (~250–300 °C). In our case, the application of 30 min of ultrasound was enough to finish the precipitation reaction, obtaining yields close to 95%.

The XRD plot in Figure 1A confirmed that the peaks of the as-synthesized powder, located at 2θ = 32.57°, 35.56°, 38.77°, 48.82°, 58.27°, 61.59°, 66.36°, and 67.92°, were assigned to the characteristic orientation planes of the pure CuO monoclinic structure [25]. The influence of sono-chemistry in the powder precipitation has avoided the presence of further undesirable species, such as brochantite and posnjakite, usually detected in works in which the same sulfate precursor was involved [23]. To analyze the effect of the ultrasonic power on the physicochemical features of the synthesized CuO NPs, crystallographic parameters were also determined. The interlaminar distance was calculated from the XRD plots, obtaining values of 3.88 Å. The crystallite size was additionally calculated according to the Warren modification of the Scherrer formula using both the peak position and the full width at half maximum (FWHM) of the (−111) reflection. The result was around 19.52 nm, which was close to other values reported in the absence of an ultrasound. Therefore, we could conclude that there are no significant differences in the final crystallographic ordering. TGA curves depicted in Figure 1B also confirmed that the as-prepared powder did not present relevant weight losses and phase conversions during a later calcination process in air. The slight decrease of mass (4%) was attributed to superficially physiosorbed and interlamellar water losses, as well as small amounts of remaining reactants.

Figure 1C illustrates the adsorption/desorption isotherm and provides details regarding the pore size distribution of the as-synthesized CuO powder calculated using the Barrett, Joyner, and Halenda (BJH) model. The isotherm profile corresponds to Type IV isotherms. The low adsorbed volume at very low relative pressure (P/Po < 0.1) confirms the limited presence of micropores, attributed to the absence of filling without adsorbent–adsorbate interactions. The CuO powder also absorbed a minimal amount of N_2_, displaying a gradual increase in the relative pressure range of 0–0.8, indicating the absence of mesopores. However, a significant amount of gas is absorbed in the relative pressure range of 0.8–1, which is characteristic of certain microporosity [26]. This increase after P/P0 > 0.9 represents the overall porosity of CuO contributed by the interspaces among nanoparticles, providing a higher exposed surface area to the surrounding medium. The pore size distributions, plotted as a function of the adsorbed volume of N_2_, confirm the presence of interspaces/macropores over a 30–70 nm range. The Specific Surface Area (SSA) of nanoparticles were finally determined using the Brunauer, Emmett, and Teller (BET) method, obtaining a value of 49.46 m^2^·g^–1^. All this info is in concordance with the morphological details revealed in the FESEM micrograph of Figure 1D. The image revealed short nanorod-like structures with a diameter of around 30 nm. Although the primary shape of particles is nanorods, the microstructural detail shows a small agglomeration degree within the mentioned interspaces among them.

To define the optimal conditions of colloidal processing, the evolution of zeta potential of as-synthesized CuO NPs was firstly examined in a wide range of pH values (2 < pH < 11), Figure 2a. These CuO NPs exhibited an isoelectric point at pH 2.5 and a negative surface charge above this value. It is remarkable that the zeta potential measurement at pH 8 was −33 mV, which is considered an idoneous condition to adsorb a cationic polyelectrolyte and improve the electrosteric effect of nanoparticles in dispersion. According to the literature [27], the branched PEI has a pKa value of 8.6, which means that their amine groups may be protonated at pH 8. The electrostatic interaction between the positive groups of PEI and the negative surface of CuO NPs was additionally followed by zeta potential analyses, Figure 2a. The negative surface of CuO NPs turned positive with the addition of 1 wt.% of PEI and achieved a maximum of +40 mV when the surfaces CuO NPs are stabilized with 6 wt.% of PEI (CuO6PEI). Further colloidal characterization, both CuO6PEI and as-prepared CNF, was carried out to check the surface charges in the liquid medium where they were then blended. The zeta potential values in function of pH have also been plotted in Figure 2b.

These results allow for selecting the appropriate pH conditions to ensure an effective electrostatic interaction between the negative charge of CNF and the positive surface of the CuO6PEI, Figure 2c. At pH 8, the zeta potential values were −45 mV and +40 mV, respectively. Therefore, all mixtures formulated from both components in a unique suspension have been made at this working pH, promoting a colloidal heterocoagulation phenomenon between them.

### 3.2. Blending Formulation and Aerogels Preparation

To define a specific ratio of the CNF/CuO6PEI formulations, different percentages of CuO6PEI were blended over the dry weight of CNF. The formation of different heterostructures was again monitored through the zeta potential evolution, Figure 3. The positively charged CuO6PEI approaches the negative sites of CNF, provoking the neutralization of its functional groups. The negative zeta potential values of bare CNF turned towards a value of 22 mV when the percentage of CuO6PEI added was 100 wt.%. Following that, the values continued increasing until a maximum of 40 mV, when the added percentage was 1500 wt.%. No further variations were observed for higher additions of CuO6PEI.

To explore the formation of the CNF/CuO6PEI heterostructures and examine the antibacterial role of the CuO6PEI, six formulations, labelled as CNF, CNF-5CuO, CNF-10CuO, CNF-25CuO, and CNF-50CuO, CNF-100CuO, were prepared by varying the percentage of added CuO-6PEI (0, 5, 10, 25, 50, and 100%). Notice that these compositions were chosen before and after turning the charge of CNF considering the antimicrobial concentrations of CuO NPs reported elsewhere [6,12,28].

After stirring, casting, and freeze-drying processes, the resulting aerogels were easily removed from the corresponding mold for further characterization. The pictures included in Figure 4a display significant changes in the physical appearance of nanocellulose-based aerogels when they are functionalized with the inorganic nanophase. All samples were slight, flexible, and easily manageable. Detachments of material were not observed. FTIR spectra were acquired to obtain information about the electrostatic interaction between CuO-6PEI and CNF, Figure 4b. The characteristic cellulosic bands were observed at 3340 cm^−1^ due to the stretching of −OH, at 2890 cm^−1^ due to aliphatic C−H stretching, at 1646 cm^−1^ due to C=O stretching of the carboxylate groups, at 1148 cm^−1^ corresponds to deformation of the antisymmetric C–O–C bond, and a broad band between 1010 and 1050 cm^−1^ was attributed to the C−OH bonds and the ether of the pyranose ring. Other peaks were additionally observed in the range of 1450–1420 cm^−1^ due to single bond CH_2_ scissoring motion in cellulose, 1368–1373 cm^−1^ represents C−H bending, ∼1317 cm^−1^ shows CH_2_ wagging, and a peak at 893 cm^−1^ with improved sharpness associated with cellulosic β-glycosidic linkages. Similarly, peaks at 1154–1159 cm^−1^ show C−C ring stretching band [29].

If the comparative analysis is concretely focused on peaks corresponding to hydroxyl and carboxyl groups (3340 and 1600 cm^−1^, respectively), a notorious decrease in their intensity was appreciated when the amount of CuO6PEI was increased, whereas the prominence of peaks corresponding to the Cu(II)-O bond (at 604 and 499 cm^−1^) presented an opposite tendency and was higher with the increase of CuO concentration.

Recent studies have demonstrated that the increase and narrowing of the band corresponding to carboxyl groups (around 1600 cm^−1^) is associated to the amount of oxidative reactant during TEMPO pre-treatment [30]. However, the decrease and widening of this peak for the same TEMPO pre-treatment conditions (similarly to this study) is associated with the electrostatic interactions between CNF and CuO6PEI. These electrostatic interactions are produced between the negative charge of the hydroxyl/carboxyl groups belonging to CNF and the positive charge of the PEI groups, surface-modifying the CuONPs. As expected, the more the CuO6PEI amount increases, the higher the electrostatic interaction is, and the higher intensity variations are.

Therefore, we can confirm the effectiveness of the colloidal heterocoagulation treatment for functionalizing the CNF, using the same oxidative capacity of the selective oxidation of cellulose.

To analyze the internal structure and microstructural details of different aerogels formulated, FESEM images were taken (Figure 5).

The typical wrinkled structure usually obtained for aerogels was herein analyzed at high magnification to appreciate the details of CuO6PEI nanoparticles over the reticulated CNF. Figure 5a, corresponding to the CNF sample without inorganic phase, has been included to make a comparison in terms of physical and textural changes.

FESEM images evidence that an increase in the ratio of the inorganic nanophase has a significant effect on the aerogel’s microstructures covering the surfaces of all aerogels. The more the CuO6PEI content increases, the lower the exposed surface of CNF.

The CNF-5CuO, CNF-10CuO, and CNF-25CuO samples with a lower inorganic content showed a high dispersion of nanoparticles, avoiding the presence of large agglomerates. The effect of the previous CuO NP dispersion/stabilization confirms the effectiveness of colloidal method when we are dealing with the blending of components for these composite materials. On the other hand, the CNF-50CuO and CNF100CuO samples with higher inorganic content showed nanoparticles overlapping. They are settled into cellulose surfaces forming interconnected layers, which are actively exposed towards the external medium, which could potentially affect their functional behavior.

### 3.3. Antimicrobial Activity of Aerogels

Results of the agar disk diffusion test are presented in Figure 6.

Overall, the antibacterial activity of the aerogels was rather effective, obtaining mean values for the widths of the antimicrobial “halos” (nwhalo) from 0 to 0.6 and 0.35 for *S. aureus* and *E. coli*, respectively. The standard deviations obtained from the mean of the three measurements (for each composition) were always between ±0.01 and 0.02.

For both bacteria, the inhibition zones were directly correlated with the amount of CuO NPs in the aerogels reaching the maximum inhibition zone with 100%-CuO NPs samples. Since copper ions must diffuse into the agar to inhibit germination and bacterial growth, the higher the amount of CuO NPs in the aerogels the more ions available to diffuse into agar, leading to larger diameters. Comparing the effect of the CuO NPs on both microorganisms, we observe a higher effectiveness of the aerogels against the gram-positive *S. aureus* compared to the gram-negative *E. coli*. This observation is also in agreement with [31,32], who reported that Gram-negative strains present more resistance/tolerance against ZnO and CuO nanoparticles over Gram-positive bacterial strains. The outer membrane (OM) present in Gram-negative bacteria acts as a protective barrier against antimicrobial agents since they must pass the outer membrane to access their targets, conferring greater resistance compared to Gram-positive bacteria [33].

Since disk diffusion method is not capable of distinguishing between bacterial growth inhibition and bacterial death, the antimicrobial kinetics of the CNF-XCuO aerogels were additionally studied by the broth dilution method. The standard deviations obtained from the mean of the two measurements (again for each composition) were ±0.25 and 0.40. Figure 7 shows the antimicrobial behavior of the samples in liquid media. Overall, the composites were able to eliminate both pathogenic bacteria in less than 5 h when CuO NPs were present in aerogels at concentrations below 50% wt.

It was noted that CNF samples caused a slight reduction in the first few hours of both studies, remaining constant thereafter. This phenomenon can be justified through different theories. On one side, an antibacterial activity of oxidized nanocellulose with different carboxyl content has been reported [34]. Additionally, since the bacteria are in liquid media, the porous structure of CNF aerogels could function as a physical impairment (network) where bacterial cells are retained, leading to their accumulation [35]. In our analyses, the cell retention was different for both bacteria, reaching reductions of around 1 log for *S. aureus* compared to 0.7 log for *E. coli*. A reasonable answer to this phenomenon could be the different features of both bacteria and the way they are arranged. First, *S. aureus* is a coccus, non-flagellated bacteria of around 1 µm diameter that is grouped in clusters [36]. In addition, *E. coli* is a rod shape, flagellated bacteria with dimensions of approximately 1 × 0.5 µm and is normally found arranged singly or in pairs [37]. The smaller size of *E. coli*, along with its active motility in liquid media, causes a lower cell retention in the aerogels compared to *S. aureus*. A clear reduction in bacterial counts was evidenced when CuO NPs were added into the samples. As shown in Figure 7a,b, the aerogels were able to deactivate *S. aureus* and *E. coli* in less than 5 h, except for CNF-50CuO samples. The size, shape, and charge of CuO NPs are crucial factors that define the ability to kill bacteria [38]. Usually, the smaller the nanoparticle the more effective the cell damage, since the surface area to volume ratio is larger as compared to bigger particles. However, the high amount of CuO NPs in CNF-50CuO samples leads to their agglomeration [13], reducing the specific surface area and therefore delaying cell death to 7 h. Notice that CNF-100CuO samples were not included as they presented worse behaviors.

Comparing the bactericidal effect for both bacteria, a greater reduction of *S. aureus* was evidenced in the first hour, reaching log reductions of 1.4, 0.99, 0.82, and 2.27 for CNF-5CuO, CNF-10CuO, CNF-25CuO, and CNF-50CuO samples, respectively. Despite the protection of *E. coli* against antimicrobial agents, a sharp decrease was observed between 1 h and 2.5 h. During the first hour of study, *E. coli* is able to place in favorable areas where the concentrations of Cu^2+^ ions in the solution are low. However, the higher release of ions from the materials to the medium at 2.5 h, together with the high cationic affinity conferred by the negatively charged lipopolysaccharide molecules present in the outer membrane of *E. coli* [38], causes concentrations to decay to those obtained for *S. aureus*.

## 4. Conclusions

-It has been demonstrated that the application of a controlled amount of ultrasonic power in a concrete volume of sulfate precursor solution (W cm^−2^ mol^−1^) allowed the instantaneous formation of a pure CuO monoclinic nanophase. The direct obtaining of this species avoided the application of further thermal treatments that are conventionally required in the absence of sonochemistry.-In terms of processing of nanocomposite materials, it can be concluded that the design of CNF–CuO heterostructures by means of a colloidal route provides numerous possibilities that are unattainable using traditional methods.-The fundamental hypothesis of the present work has been confirmed. The TEMPO-Oxidized CNF showed idoneous surface charge to promote their electrostatic interaction with highly dispersed CuO nanostructures. It was observed that with increasing CuO concentration in the formulation, the CNF surface was completely covered, increasing the exposition of the inorganic phase towards the media and the corresponding improvement in the functional performance of the final aerogels.-All samples showed a positive response against Escherichia coli and Staphylococcus aureus. An increase of the antibacterial response of the aerogels, measured by agar disk diffusion test, was observed with the increase of CuO NPs incorporated, obtaining a width of the antimicrobial “halo” (nw_halo_) from 0 to 0.6 and 0.35 for *S. aureus* and *E. coli*, respectively. Furthermore, the aerogels were also able to deactivate *S. aureus* and *E. coli* in less than 5 h when the antibacterial assays were analyzed by a broth dilution method.-It is therefore concluded that the use of CNF as a support for antimicrobial materials favors the adequate dispersion and support of the nanoparticles, and its aerogel conformation and specific cross-linked structure facilitate good contact and transfer towards different media, increasing the efficiency of the antimicrobial activity. Although this work involved the incorporation of CuO NPs into a specific nanocellulose-based support through cross-linking phenomena, the colloidal procedure, herein described, opens a door to ad hoc design of multifunctional membranes, filters, or flexible substrates, which requires the inclusion of other alternative compositions.

## Figures and Tables

**Figure 1 bioengineering-10-01312-f001:**
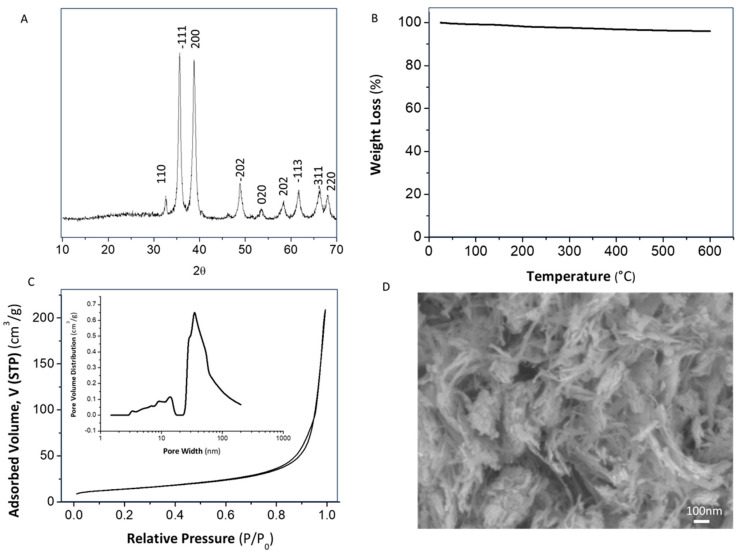
Characterization of as-synthetized CuONP: (**A**) XRD pattern, (**B**) TGA curve, (**C**) N_2_ adsorption/desorption isotherm and pore size distribution, and (**D**) FESEM micrographs.

**Figure 2 bioengineering-10-01312-f002:**
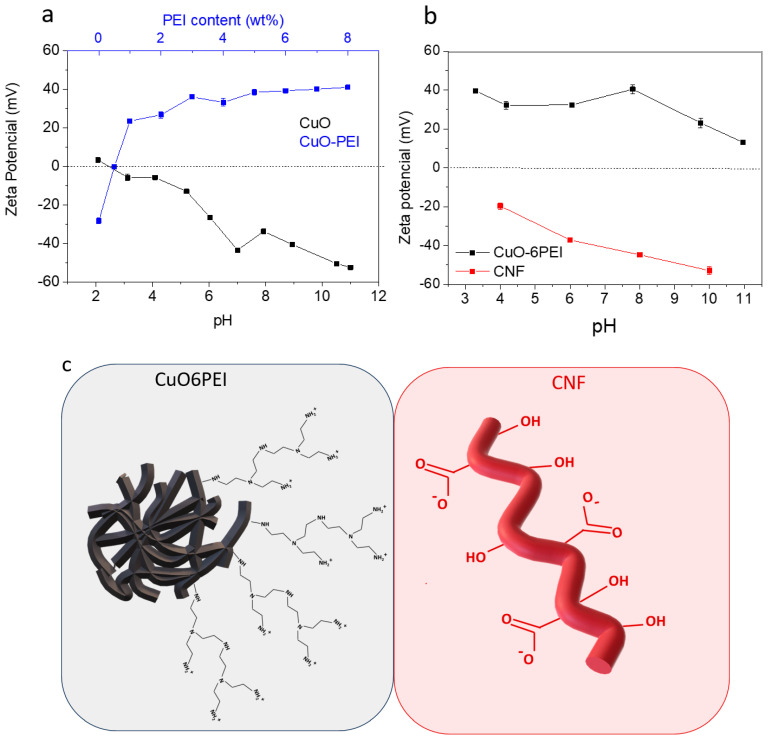
Evolution of Zeta potential as a function of amount of adsorbed PEI onto the particle surface and evolution of Zeta potential of CuO as a function of pH without PEI (**a**); Evolution of Zeta potential of CuO–6PEI and CNF as a function of pH (**b**); Scheme of the involved component representing their surface charge (**c**).

**Figure 3 bioengineering-10-01312-f003:**
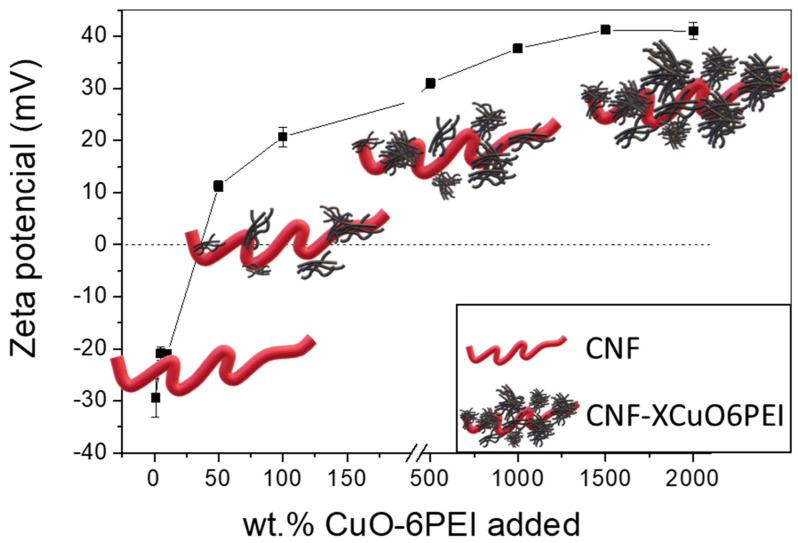
Evolution of Zeta potential as a function of the amount of adsorbed CuO-6PEI onto the CNF. The scheme of the figure represents the CNF in red and the XCuO6PEI in black.

**Figure 4 bioengineering-10-01312-f004:**
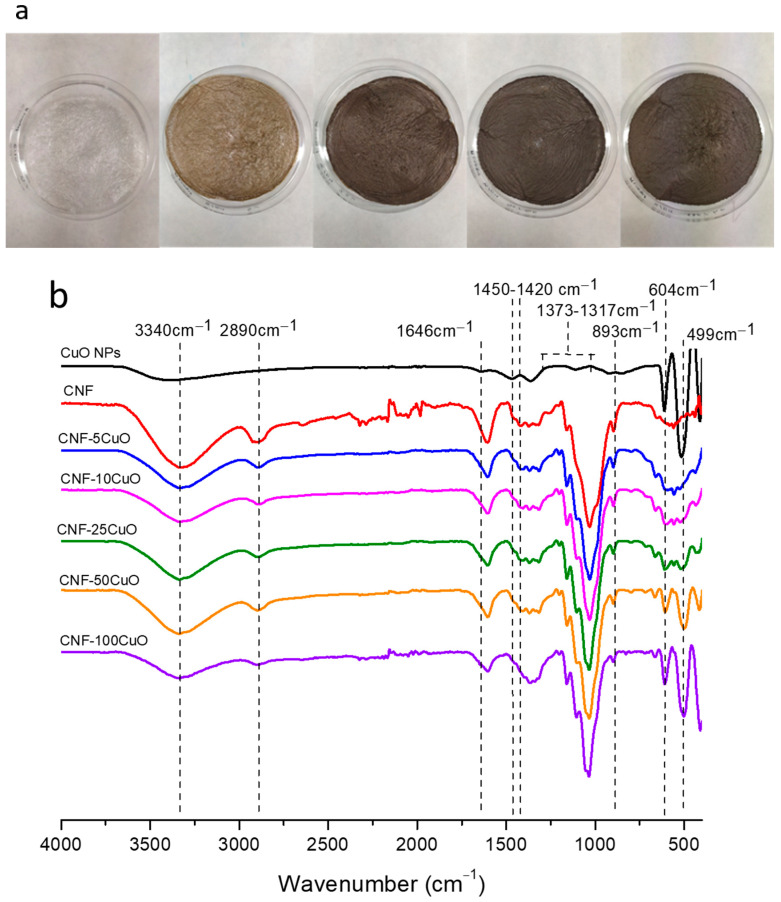
(**a**) Picture and (**b**) FTIR spectra of CNF-XCuO6PEI aerogels.

**Figure 5 bioengineering-10-01312-f005:**
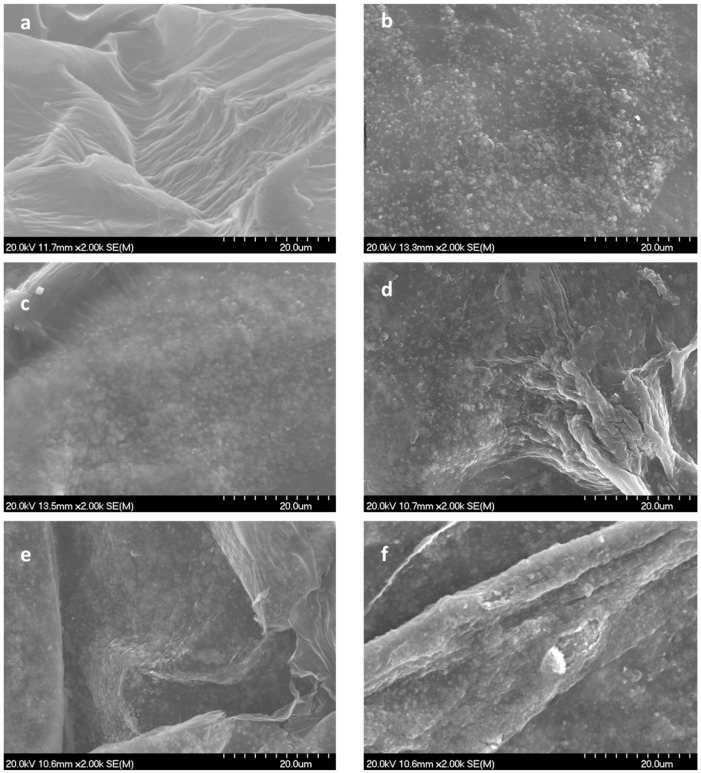
Scanning electron micrographs of a fractured (top view) aerogel. Letters (**a**–**f**) correspond to CNF, CNF-5CuO, CNF-10CuO, CNF-25CuO, CNF-50CuO, and CNF-100CuO, respectively.

**Figure 6 bioengineering-10-01312-f006:**
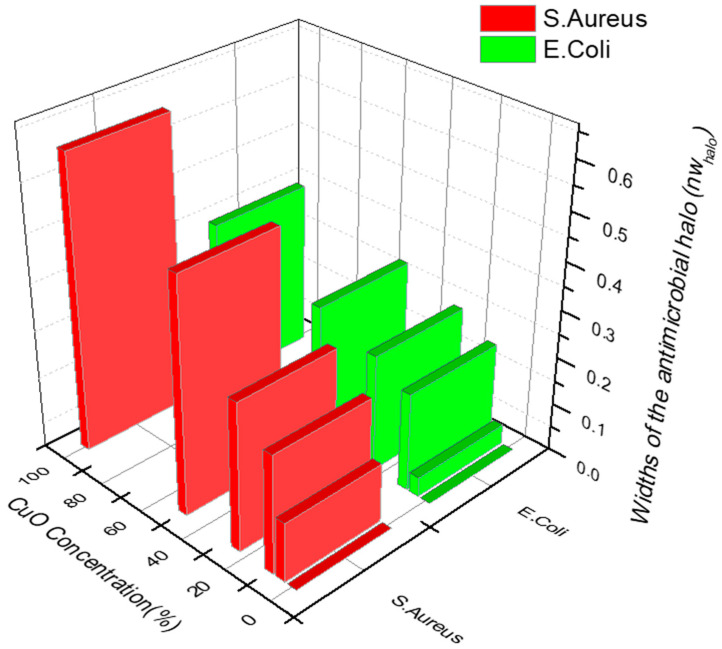
Antibacterial activity against *E. coli* and *S. aureus* by the Agar Diffusion method.

**Figure 7 bioengineering-10-01312-f007:**
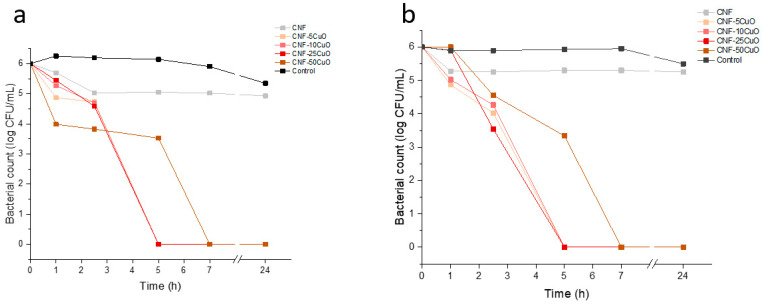
Antibacterial activity of CNF/CuO aerogels against *S. aureus* (**a**) and *E. coli* (**b**).

## Data Availability

Data available on request.

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
