# Peer review of "Antibacterial Aerogels-Based Membranes by Customized Colloidal Functionalization of TEMPO-Oxidized Cellulose Nanofibers Incorporating CuO"

_bioengineering, 2023, doi:10.3390/bioengineering10111312_

Round 1

Reviewer 1 Report

Comments and Suggestions for Authors

This paper reports the fabrication of composites consisting of cellulose nanofibers coated with CuO nanoparticles in the form of nanorods and their use as antibacterial aerogels. The formulation of the composites was performed based on the colloidal heterocoagulation treatment already described previously with other inorganic nanoparticles. I believe that the paper is good as it is and I do not have any further comments to make. I recommend it for publication.

Author Response

We acknowledge the reviewer time for revising our work and we thank him/her for accepting it in the current form.

Reviewer 2 Report

Comments and Suggestions for Authors

Paper devoted to investigation of antibacterial aerogels-based membranes by customized colloidal functionalization of TEMPO-oxidized cellulose nanofibers incorporating CuO. This topic is relevant and interesting. Within the framework of this research the involved components of formulated blending, CNF and CuO NPs, were individually obtained by using a biorefinery strategy for agricultural waste valorization together with an optimized chemical precipitation, assisted by ultrasounds. The optimization of synthesis parameters for CuO NPs have avoided the presence of undesirable species which usually requires of later thermal treatment with associated costs. The aerogels-based structure, obtained by conventional freeze-drying, acted as 3D support for CuO NPs providing a good dispersion within the cross-linked structure of the nanocellulose and facilitating direct contact of the antibacterial phase against undesirable microorganisms. All samples showed a positive response against Escherichia coli and Staphylococcus aureus. An increase of the antibacterial response of the aerogels, measured by agar disk diffusion test, has been observed with the increase of CuO NPs incorporated, obtaining width of the antimicrobial “halo” (nwhalo) from 0 to 0.6 and 0.35 for S. aureus and E. coli, respectively. Furthermore, the aerogels have been able to deactivate S. aureus and E. coli in less than 5 hours when the antibacterial assays have been analyzed by broth dilution method. From CNF-50CuO samples, an overlap nanoparticles effect produced a decrease of the antimicrobial kinetic.

This paper is well organized and good written. But there is a comment. I think that the Сonclusions should be divided into several points. This will improve their perception by readers.

Author Response

Thank you to the reviewer comment. The conclusion section has been modified according to his/her suggestion. The highlighted manuscript includes the changes.

Reviewer 3 Report

Comments and Suggestions for Authors

In this study, the authors synthesized CuO nanoparticles (NPs) via a sonochemical approach. The crystallography and porosity of the as-synthesized CuO NPs were characterized using XRD, adsorption/desorption isotherms, and FESEM analyses. Subsequently, the zeta potentials of CuO, CuO-PEI, and cellulose nanofibers (CNF) were measured across different pH values to get the optimal conditions for colloidal processing. CNF/CuO-PEI aerogels were then fabricated by freeze-drying at various CuO-PEI to CNF weight ratios. The morphologies of these aerogels were further examined using FESEM. Lastly, the antibacterial efficacy of the aerogels was evaluated against S. aureus and E. coli using both dish diffusion and broth dilution methods. Following the resolution of minor concerns, I am of the opinion that the work is well-suited for publication in Bioengineering.

1.     Page 2, line 83, typo of “metal noble”, should be noble metal.

2.     Page 5, line 238, typo of “Figure 2a”, should be “Figure 1b”.

3.     Page 5, line 254, please provide the full name for the abbreviation “SSA”. I assume it means specific surface area, but could be clear for authors to introduce the full name.

4.     Page 6, please center the Figure 1.

5.     Page 12, line 417 and 418, typo on capitalization of the “Oxidized Nanocellulose” and “Carboxyl”.

Author Response

Thank you for the nice comments. We have included some modifications in text according to the following suggestions.

Page 2, line 83, typo of “metal noble”, should be noble metal.

The mistake has been corrected.

Page 5, line 238, typo of “Figure 2a”, should be “Figure 1b”.

The mistake has been corrected.

Page 5, line 254, please provide the full name for the abbreviation “SSA”. I assume it means specific surface area but could be clear for authors to introduce the full name.

The abbreviation has been clarified in text.

Page 6, please center the Figure 1.

The figure has been centered.

Page 12, line 417 and 418, typo on capitalization of the “Oxidized Nanocellulose” and “Carboxyl”

The text has been corrected.

Reviewer 4 Report

Comments and Suggestions for Authors

The authors present inovative components by formulated blending, CNF and CuO NPs, were individually obtained by using a biorefinery strategy for agricultural waste valorization together with an optimized chemical precipitation, assisted by ultrasounds. The optimization of synthesis parameters for CuO NPs have avoided the presence of undesirable species which usually requires of later thermal treatment with associated costs. Also, the aerogels-based structure, obtained by conventional freeze-drying, acted as 3D support for CuO NPs providing a good dispersion within the cross-linked structure of the nanocellulose and facilitating direct contact of the antibacterial phase against undesirable microorganisms. 

Minor comments:

1. Why are the deviations under figures 6 and 7 not shown?

2. In part matherials and methods , p 2.2. to be rivised.

3. The passage from lines 223 to 241 is too descriptive, please shorten and rework it.

4. I propose to introduce a chapter limitation of the study.

5. The part conclusions to be rivised.

6.Some of the references do not meet the requirements of the journal.

7.I suggest that the authors add an introductory sheme after the abstract, this will make the innovation of the study realy visible. 

Comments on the Quality of English Language

 Minor editing of English language required

Author Response

We thank the revision of the referee and try to give a good response to his/her minor comments as follow:

Minor comments:

  1. Why are the deviations under figures 6 and 7 not shown?

Three and two measurements of each composition were carried out to perform the agar disk diffusion and broth dilution tests, respectively.  Further information about the calculations of the means and the standard deviations have been included in methodology and results sections.

  1. In part materials and methods, p 2.2. to be revised.

The methodology section has been revised to clarify the point 2.2

  1. The passage from lines 223 to 241 is too descriptive, please shorten and rework it.

The paragraph has been shortened and partially modified. Nevertheless, the information related to crystallographic parameters has been kept according to the comment of another reviewer.

  1. I propose to introduce a chapter limitation of the study.

Although the proposal of adding a chapter limitation could clarify partial data to some potential reader in other works more generalist, we consider that our manuscript already includes several details with similar information. Moreover, this journal does not usually incorporate this type of information as a separate section. Therefore, we prefer to keep the text in the current form.

  1. The part conclusions to be revised.

The conclusion section has been modified according to his/her suggestion. The highlighted manuscript includes the changes.

  1. Some of the references do not meet the requirements of the journal.

The references section has been revised.

     7.I suggest that the authors add an introductory scheme after the abstract,           this will make the innovation of the study really visible. 

A graphical abstract has been incorporated in the new version.

Reviewer 5 Report

Comments and Suggestions for Authors

E. Usala, E. Espinosa, W. El Arfaoui, R. Morcillo, B. Ferrari and Zoilo Gonzalez: Antibacterial Aerogels-based Membranes by customized Colloidal Functionalization of TEMPO-Oxidized Cellulose Nanofibers incorporating CuO

The manuscript is written well, and it can be followed easily. The findings are described correctly, and the conclusions can be accepted, mostly. Problems can only be found in formal details.

My remarks and suggestions are as follows.

Using the Scherrer formula (even its modified version), the size of the coherent domains can be obtained. The nanoparticles contain more domains than one, therefore the particle size is much greater than the value deduced by the equation. (Small angle scattering can provide correct values.)

In Fig. the conditions of the adsorbed gas volume must be written (for example STP, standard temperature and pressure). Only reasonable accuracy of measured values must be given (49.46 m2/g). In SEM pictures the size-bar is missing, or cannot see well (Fig. 1., 5).

The text and the Fig. 4 (b) should match with each other. Only four bands are designed in the figure, but the other not.

A more detailed description is required in Fig. 4 (a).

In Fig. 5, the SEM micrograph shows a smooth surface morphology (a), but in the next picture (b) the surface is covered by great number of small grains (a very course surface appears). It does not fit somehow into the changes of the surface.

I suggest improving the visual appearance of Fig.7.

I recommend brushing through the English of the manuscript. (fruitful > rather affective, The pH of work was around 8 > the pH of the solution, used in the present work, was ..)

Author Response

We thank the reviewer the nice comments about our work. We try to complete de manuscript according to all proposed suggestions.

-Using the Scherrer formula (even its modified version), the size of the coherent domains can be obtained. The nanoparticles contain more domains than one, therefore the particle size is much greater than the value deduced by the equation. (Small angle scattering can provide correct values.)

Although the Small Angle Scattering can provide much more information about the domains of synthetized CuO nanoparticles, the study of their crystallographic parameters is not the main aim of this work. We thank the referee suggestion, and we consider the idea to increase the info in the forthcoming future.

-In Fig 1C. the conditions of the adsorbed gas volume must be written (for example STP, standard temperature and pressure). Only reasonable accuracy of measured values must be given (49.46 m2/g). In SEM pictures the size-bar is missing, or cannot see well (Fig. 1., 5).

Figure 1C and 1D has been modified according to the referee comment. SEM micrograph of Figure 5 has been increased to clarify the view of the size-bar.

-The text and the Fig. 4 (b) should match with each other. Only four bands are designed in the figure, but the other not.

All bands in figure 4b has been matched with text

-A more detailed description is required in Fig. 4 (a).

Further information about the pictures has been included in text

-In Fig. 5, the SEM micrograph shows a smooth surface morphology (a), but in the next picture (b) the surface is covered by great number of small grains (a very course surface appears). It does not fit somehow into the changes of the surface.

A new sentence has been included in the manuscript to clarify this misunderstanding. Concretely, the figure 5a corresponds to the CNF sample without inorganic nanophase. This is the main reason to appreciate the textural changes among micrographs.

-I suggest improving the visual appearance of Fig.7.

The size of figure has been increased in order to visualize the data in a better way

-I recommend brushing through the English of the manuscript. (fruitful > rather effective, The pH of work was around 8 > the pH of the solution, used in the present work, was ..)

These expressions and other have been modified along the manuscript.
